# Selectively Sharing Experiences Improves Multi-Agent Reinforcement Learning

**Matthias Gerstgrasser**
School of Engineering and Applied Sciences
Harvard University
Computer Science Department
Stanford University
matthias@seas.harvard.edu

**Tom Danino**      **Sarah Keren**
The Taub Faculty of Computer Science
Technion - Israel Institute of Technology
tom.danino@campus.technion.ac.il
sarahk@cs.technion.ac.il

## Abstract

We present a novel multi-agent RL approach, *Selective Multi-Agent Prioritized Experience Relay*, in which agents share with other agents a limited number of transitions they observe during training. The intuition behind this is that even a small number of relevant experiences from other agents could help each agent learn. Unlike many other multi-agent RL algorithms, this approach allows for largely decentralized training, requiring only a limited communication channel between agents. We show that our approach outperforms baseline no-sharing decentralized training and state-of-the art multi-agent RL algorithms. Further, sharing only a small number of highly relevant experiences outperforms sharing all experiences between agents, and the performance uplift from selective experience sharing is robust across a range of hyperparameters and DQN variants.

## 1   Introduction

Multi-Agent Reinforcement Learning (RL) is often considered a hard problem: The environment dynamics and returns depend on the joint actions of all agents, leading to significant variance and non-stationarity in the experiences of each individual agent. Much recent work [29, 23] in multi-agent RL has focused on mitigating the impact of these. Our work views multi-agent RL more positively, by treating the presence of other agents can also be an asset that can be leveraged. In particular, we show that groups of agents can collaboratively explore the environment more quickly than individually.

To this end, we present a novel multi-agent RL approach that allows agents to share a small number of experiences with other agents. The intuition is that if one agent discovers something important in the environment, then sharing this with the other agents should help them learn faster. However, it is crucial that only important experiences are shared - we will see that sharing all experiences indiscriminately will not improve learning. To this end, we make two crucial design choices: Selectivity and priority. Selectivity means we only share a small fraction of experiences. Priority is inspired by a well-established technique in single-agent RL, *prioritized experience replay* (PER) [30]. With PER an off-policy algorithm such as DQN [24] will sample experiences not uniformly, but proportionally to "how far off" the current policy's predictions are in each state, formally the *temporal difference (td) error* . We use this same metric to prioritize which experiences to share with other agents.

We dub the resulting multiagent RL approach "Selective Multi-Agent Prioritized Experience Relay" or "SUPER". In this, agents independently use a DQN algorithm to learn, but with a twist: Each agent relays its highest-td-error experiences to the other agents, who insert them directly into their replay buffer, which they use for learning. This approach has several advantages:

37th Conference on Neural Information Processing Systems (NeurIPS 2023).

1. It consistently leads to faster learning and higher eventual performance, across hyperparameter settings.

2. Unlike many "centralized training, decentralized execution" approaches, the SUPER learning paradigm allows for (semi-) decentralized training, requiring only a limited bandwidth communication channel between agents.

3. The paradigm is agnostic to the underlying decentralized training algorithm, and can enhance many existing DQN algorithms. SUPER can be used together with PER, or without PER.[1]

In addition to the specific algorithm we develop, this work also introduces the paradigm of "decentralized training with communication". This is a 'middle ground between established approaches of decentralized and centralized training (including "centralized training, decentralized execution").

In the remainder of the paper, we will discuss related literature and technical preliminaries; introduce our novel algorithm in detail; describe experimental evaluation and results; and suggest avenues for future work.

## 2   Related Work

**Multi-Agent RL** approaches in the literature can be broadly categorized according to the degree of awareness of each learning agent of the other agents in the system [37]. On one end of this spectrum is independent or decentralized learning where multiple learning agents (in the same environment) are unaware or agnostic to the presence of other agents in the system [36, 19]. From the perspective of each agent, learning regards the other agents as part of the non-stationary environment. At the opposing end of the spectrum, in centralized control a single policy controls all agents. In between these two, a number of related strands of research have emerged.

Approaches nearer the decentralized end of the spectrum often use communication between the agents [46, 9, 13]. Several forms of cooperative communication have been formulated, by which agents can communicate various types of messages, either to all agents or to specific agent groups through dedicated channels [17, 33, 26, 27]. While communication among agents could help with coordination, training emergent communication protocols also remains a challenging problem; recent empirical results underscore the difficulty of learning meaningful emergent communication protocols, even when relying on centralized training [16]. Related to this, and often used in conjunction, is modeling other agents [2, 16, 10, 44], which equips agent with some model of other agent's behavior.

This in turn is related to a number of recent approaches using centralized training but decentralized execution. A prevailing paradigm within this line of work assumes a training stage during which a shared network (such as a critic) can be accessed by all agents to learn decentralised (locally-executable) agent policies [23, 1, 28, 10, 29]. These approaches successfully reduce variance during training, e.g. through a shared critic accounting for other agents' behavior, but rely on joint observations and actions of all agents. Within this paradigm, and perhaps closest related to our own work, [7] introduces an approach in which agents share (all of their) experiences with other agents. While this is based on an on-policy actor-critic algorithm, it uses importance sampling to incorporate the off-policy data from other agents, and the authors show that this can lead to improved performance in sparse-reward settings. Our work also relies on sharing experiences among agents, but crucially relies on selectively sharing only some experiences. [2]

A separate but related body of work is on transfer learning [8]. While this type of work is not directly comparable, conceptually it uses a similar idea of utilizing one agent's learning to help another agent.

**Off-Policy RL** The approach we present in this paper relies intrinsically on off-policy RL algorithms. Most notable in this class is DQN [24], which achieved human-level performance on a wide variety of Atari 2600 games. Various improvements have been made to this algorithm since then, including

---

[1] Note that while the name "SUPER" pays homage to PER, SUPER is not related to PER other than using the same heuristic for relevance of experiences.

[2] Interestingly, in the appendix to the arXiv version of that paper, the authors state that experience sharing in DQN did not improve performance in their experiments, in stark contrast with the result we will present in this paper. We believe this may be because their attempts shared experiences indiscriminately, compared to our selective sharing approach. We will see that sharing all experiences does not improve performance as much or as consistently as selective sharing.

dueling DQN [42], Double DQN (DDQN) [41], Rainbow [14] and Ape-X [15]. Prioritized Experience Replay [30] improves the performance of many of these, and is closely related to our own work. A variant for continuous action spaces is DDPG [32]. Off-policy actor-critic algorithms [12, 21] have also been developed, in part to extend the paradigm to continuous control domains.

## 3    Preliminaries

**Reinforcement learning (RL)** deals with sequential decision-making in unknown environments [34]. *Multi-Agent Reinforcement Learning* extends RL to multi-agent settings. A common model is a *Markov game*, or *stochastic game* defined as a tuple $\langle S, \mathcal{A}, \mathcal{R}, \mathcal{T}, \gamma \rangle$ with states $S$, *joint actions* $\mathcal{A} = \{A^i\}_{i=1}^{n}$ as a collection of action sets $A^i$, one for each of the $n$ agents, $\mathcal{R} = \{R^i\}_{i=1}^{n}$ as a collection of reward functions $R^i$ defining the reward $r^i(a_t, s_t)$ that each agent receives when the joint action $a_t \in \mathcal{A}$ is performed at state $s_t$, and $\mathcal{T}$ as the probability distribution over next states when a joint action is performed. In the partially observable case, the definition also includes a joint observation function, defining the observation for each agent at each state. In this framework, at each time step an agent has an *experience* $e = < S_t, A_t, R_{t+1}S_{t+1} >$, where the agent performs action $A_t$ at state $S_t$ after which reward $R_{t+1}$ is received and the next state is $S_{t+1}$. We focus here on decentralized execution settings in which each agent follows its own individual policy $\pi_i$ and seeks to maximize its discounted accumulated return. The algorithm we propose in this paper further requires that the Markov game is "anonymous", meaning all agents have the same action and observation spaces and the environment reacts identically to their actions. This is needed so that experiences from one agent are meaningful to another agent as-is. Notice however that this does not imply that it is necessarily optimal for all agents to adopt identical behavior.

Among the variety of RL solution approaches [35, 34], we focus here on *value-based methods* that use state and action estimates to find optimal policies. Such methods typically use a value function $V_\pi(s)$ to represent the expected value of following policy $\pi$ from state $s$ onward, and a *Q-function* $Q_\pi(s, a)$, to represent for a given policy $\pi$ the expected rewards for performing action $a$ in a given state $s$ and following $\pi$ thereafter.

Q-learning [43] is a *temporal difference* (td) method that considers the difference in $Q$ values between two consecutive time steps. Formally, the update rule is $Q(S_t, A_t) \leftarrow Q(S_t, A_t) + \alpha[R_{t+1} + \gamma \max_{a'} Q(S_{t+1}, a') - Q(S_t, A_t))]$, where, $\alpha$ is the *learning rate* or *step size* setting the amount by which values are updated during training. The learned action-value function, Q, directly approximates $q^*$, the optimal action-value function. During training, in order to guarantee convergence to an optimal policy, Q-learning typically uses an $\epsilon$-greedy selection approach, according to which the best action is chosen with probability $1 - \epsilon$, and a random action, otherwise.

Given an experience $e_t$, the *td-error* represents the difference between the Q-value estimate and actual reward gained in the transition and the discounted value estimate of the next best actions. Formally,

$$\text{td}(e_t) = |R + \gamma \max_{a'} Q(S', a') - Q(S, A)| \tag{1}$$

In the vanilla version of Q-learning Q-values are stored in a table, which is impractical in many real-world problem due to large state space size. In deep Q-Learning (DQN) [25], the Q-value table is replaced by a function approximator typically modeled using a neural network such that $Q(s, a, \theta) \approx Q^*(s, a)$, where $\theta$ denotes the neural network parameters.

**Replay Buffer (RB) and Prioritized RB:** An important component in many off-policy RL implementations is a replay buffer [22], which stores past experiences. Evidence shows the replay buffer to stabilize training of the value function for DQN [24, 25] and to reduce the amount of experiences required for an RL-agent to complete the learning process and achieve convergence [30].

Initial approaches that used a replay buffer, uniformly sampled experiences from the buffer. However, some transitions are more effective for the learning process of an RL agents than others [31]. *Prioritized Experience Replay (PER)* [30] explores the idea that replaying and learning from some transitions, rather than others, can enhance the learning process. PER suggests replacing the standard sampling method, where transitions are replayed according to the frequency they were collected from the environment, with a td-error based method, where transitions are sampled according to the value of their td-error.

As a further extension, **stochastic prioritization** balances between strictly greedy prioritization and uniform random sampling. Hence, the probability of sampling transition $i$ is defined as:

$$P(i) = \frac{p_i^\alpha}{\sum_k p_k^\alpha} \qquad (2)$$

where $p_i > 0$ is the priority associated to transition $i$ and $\alpha$ determines the weight of the priority ($\alpha = 0$ is uniform sampling). The value of $p_i$ is determined according to the magnitude of the td-error, such that $p_i = |\delta_i| + \epsilon$, $\delta_i$ is the td-error and $\epsilon$ is a small positive constant that guarantees that transitions for which the td-error is zero have a non-zero probability of being sampled from the buffer.

## 4  SUPER: Selective Multi-Agent Prioritized Experience Relay

Our approach is rooted in the same intuition as PER: that not all experiences are equally relevant. We use this insight to help agents learn by sharing between them only a (small) number of their most relevant experiences. Our approach builds on standard DQN algorithms, and adds this experience sharing mechanism between collecting experiences and performing gradient updates, in each iteration of the algorithm:

1. (DQN) Collect a rollout of experiences, and insert each agent's experiences into their own replay buffer.
2. (SUPER) Each agent shares their most relevant experiences, which are inserted into all the other agents' replay buffers.
3. (DQN) Each agent samples a minibatch of experiences from their own replay buffer, and performs gradient descent on it.

Steps 1 and 3 are standard DQN; we merely add an additional step between collecting experiences and learning on them. As a corollary, this same approach works for any standard variants of steps 1 and 3, such as dueling DQN [42], DDQN [41], Rainbow [14] and other DQN improvements. Algorithm 1 in the appendix gives a more detailed listing of this algorithm. Notice that the only interaction between the agents training algorithms is in the experience sharing step. This means that the algorithm can easily be implemented in a decentralized manner with a (limited) communications channel.

### 4.1  Experience Selection

We describe here three variants of the SUPER algorithm, that differ in how they select experiences to be shared.

**Deterministic Quantile experience selection**   The learning algorithm keeps a list $l$ of the (absolute) td-errors of its last $k$ experiences ($k = 1500$ by default). For a configured bandwidth $\beta$ and a new experience $e_t$, the agent shares the experience if its absolute td-error $|\text{td}(e_t)|$ is at least as large as the $k * \beta$-largest absolute td-error in $l$. In other words, the agent aims to share the top $\beta$-quantile of its experiences, where the quantile is calculated over a sliding window of recent experiences.

$$|\text{td}(e_t)| \geq \text{quantile}_\beta(\{e_{t'}\}_{t'=t-k}^t)$$

**Deterministic Gaussian experience selection**   In this, the learning algorithm calculates the mean $\mu$ and variance $\sigma^2$ of the (absolute) td-errors of the $k$ most recent experiences ($k = 1500$ by default). It then shares an experience $e_t$ if

$$|\text{td}(e_t)| \geq \mu + c \cdot \sigma^2 \qquad (3)$$

where $c$ is a constant chosen such that $1 - \text{cdf}_\mathcal{N}(c) = \beta$. In other words, we use the $c$-quantile of a normal distribution with the (sample) mean and variance of most recent experiences. We include and benchmark this variant for two reasons. One, intuitively, we might want to be more sensitive to clusters of outliers, where using a quantile of the actual data might include only part of the cluster, while a Gaussian model might lead to the entire cluster being included. Two, mean and variance could be computed iteratively without keeping a buffer of recent td-errors, and thereby reducing memory requirements. We aim to benchmark if this approximation impacts performance.

**Stochastic weighted experience selection** Finally, and most closely related to classical single-agent PER, this variant shares each experience with a probability that's proportional to its absolute td-error. In PER, given a train batch size $b$, we sample $b$ transitions from the replay buffer without replacement, weighted by each transition's td-error. In SUPER, we similarly aim to sample a $\beta$ fraction of experiences, weighted by their td-errors. However, in order to be able to sample transitions online, we calculate for each experience individually a probability that approximates sampling-without-replacement in expectation. Formally, taking $p_i = |\text{td}(e_i)|$, we broadcast experience $e_t$ with probability

$$p = \min\left(1, \beta \cdot \frac{p_i^\alpha}{\sum_k p_k^\alpha}\right)$$

similarly to equation 2, and taking the sum over a sliding window over recent experiences. It is easy to see that if $\beta = 1/\text{batchsize}$, this is equivalent to sampling a single experience, weighted by td-error, from the sliding window. For larger bandwidth $\beta$, this approximates sampling multiple experiences without replacement: If none of the $\beta \cdot \frac{p_i^\alpha}{\sum_k p_k^\alpha}$ terms are greater than 1, this is exact. If we have to truncate any of these terms, we slightly undershoot the desired bandwidth.

In our current experiments, we share experiences and update all quantiles, means and variances once per sample batch, for conveniendce and performance reasons; However, we could do both online, i.e. after every sampled transition, in real-world distributed deployments. We do not expect this to make a significant difference.

## 5 Experiments and Results

We evaluate the SUPER approach on a number of multiagent benchmark domains.

### 5.1 Algorithm and control benchmarks

**Baseline: DDQN** As a baseline, we use a fully independent dueling DDQN algorithm. The algorithm samples a sequence of experiences using joint actions from all the agents' policies, and then inserts each agent's observation-action-reward-observation transitions into that agent's replay buffer. Each agent's policy is periodically trained using a sample from its own replay buffer, sampled using PER. We refer to this baseline as simply "DDQN" in the remainder of this section and in figures. In Section C in the appendix we also discuss a variant based on vanilla DQN.

**SUPER-DDQN** We implement SUPER on top of the DDQN baseline. Whenever a batch of experiences is sampled, each agent shares only its most relevant experiences (according to one of the criteria described in the previous section) which are then inserted into all other agents' replay buffers. As above, we run the SUPER experience sharing on whole sample batches. All DQN hyperparameters are unchanged - the only difference from the baseline is the addition of experience sharing between experience collection and learning. This allows for controlled experiments with like-for-like comparisons.

**Parameter-Sharing DDQN** In parameter-sharing, all agents share the same policy parameters. Note that this is different from joint control, in which a single policy controls all agents simultaneously; In parameter sharing, each agent is controlled independently by a copy of the policy. Parameter sharing has often been found to perform very well in benchmarks, but also strictly requires a centralized learner [40, 39, 11]. We therefore do not expect SUPER to outperform parameter sharing, but include it for completeness and as a "best case" fully centralized comparison.

**Multi-Agent Baselines** We further compare against standard multi-agent RL algorithms, specifically MADDPG [23] and SEAC [7]. Like parameter-sharing, these are considered "centralized training, decentralized execution" approaches.

### 5.2 Environments

As discussed in the Preliminaries (Section 3), SUPER applies to environments that are anonymous. We are furthermore particularly interested in environments that have a separate reward signal for

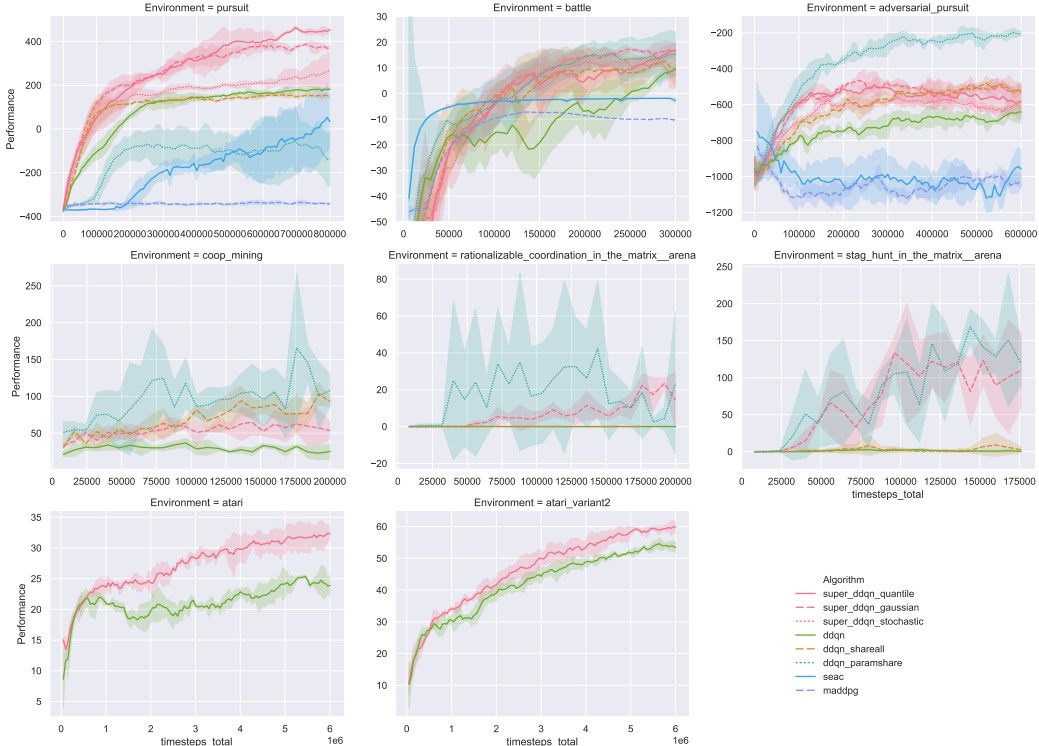

Figure 1: Performance of SUPER-dueling-DDQN variants with target bandwidth 0.1 on all domains. For team settings, performance is the total reward of all agents in the sharing team; for all other domains performance is the total reward of all agents. Shaded areas indicate one standard deviation.

each agent.[3] We therefore run our experiments on several domains that are part of well-established benchmark packages. These include three domains from the PettingZoo package [38], three domains from the MeltingPot package [18], and a two-player variant of the Atari 2600 game Space Invaders. We describe these domains in more detail in Section B in the appendix. We run a particularly large number of baselines and ablations in the PettingZoo domains, and focus on the most salient comparisons in the MeltingPot and Atari domains within available computational resources. Pursuit is particularly well suited to showcase our algorithm, as it strictly requires multiple agents to cooperate in order to generate any reward, and each cooperating agent needs to perform slightly different behavior (tagging the target from a different direction).

## 5.3 Experimental Setup

In Pursuit, we train all agents concurrently using the same algorithm, for each of the algorithms listed. Note that only the pursuers are agents in this domain, whereas the evaders move randomly. In the standard variants of Battle and Adversarial-Pursuit, we first pre-train a set of agents using independent dueling DDQN, all agents on both teams being trained together and independently. We then take the pre-trained agents of the opposing team (red team in Battle, predator team in Adversarial-Pursuit), and use these to control the opposing team during main training of the blue respectively prey team. In this main training phase, only the blue / prey team agents are trained using each of the SUPER and benchmark algorithms, whereas the red / predator team are controlled by the pre-trained policies with no further training. Figure 1 shows final performance from the main training phase. In Battle

---

[3]In principle, SUPER could also be used in shared-reward environments, but these present an additional credit assignment problem. While SUPER could potentially be combined with techniques for solving that problem such as QMIX [29], the additional complexity arising from this would make such experiments less meaningful.

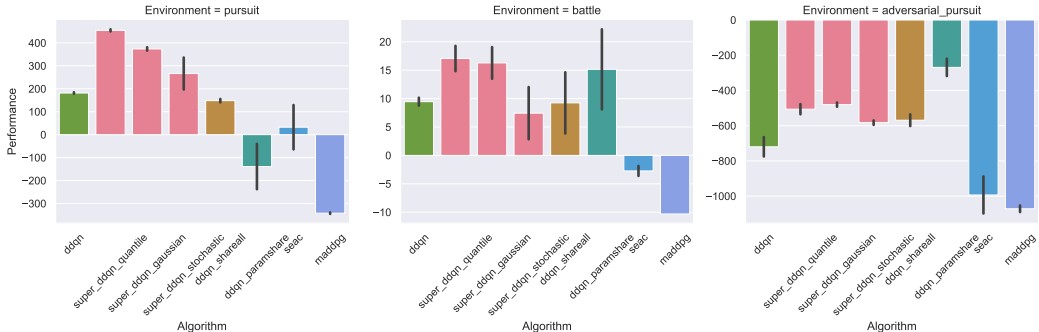

Figure 2: Performance of SUPER-dueling-DDQN variants and baselines on all three PettingZoo domains. For Pursuit, performance is the total mean episode reward from all agents. For Battle and Adversarial-Pursuit, performance is the total mean episode reward from all agents in the sharing team (blue team in Battle, prey team in Adversarial-Pursuit). Shaded areas indicate one standard deviation.

and Adversarial-Pursuit we further tested a variant where the opposing team are co-evolving with the blue / prey team, which we discuss in Section C in the appendix.

## 5.4 Performance Evaluation

Figure 1 shows performance of SUPER implemented on dueling DDQN ("SUPER DDQN") compared to the control benchmarks discussed above. In addition, Figure 2 shows final performance in the three PettingZoo environments in a barplot for easier readability. In summary, we see that (quantile and gaussian) SUPER (red bars) clearly outperform baseline DDQN on all domains, as well as both SEAC and MADDPG baselines where we were able to run them. Particularly notable is a jump in performance from DDQN (green) to quantile SUPER (leftmost red bars in Figure 2) in the PettingZoo domains. Performance of SUPER is even comparable to that of parameter sharing, our "best case" fully centralized comparison benchmark. Aside from parameter sharing, SUPER wins or ties all pairwise comparisons against baselines. Results further hold for additional variations of environments with co-evolving opponents, when using SUPER with a plain (non-dueling, non-double) DQN algorithm, and across hyperparameter settings, which we discuss in the appendix.

In more detail, we find that SUPER (red) consistently outperforms the baseline DQN/DDQN algorithm (green), often significantly. For instance in Pursuit SUPER-DDQN achieves over twice the reward of no-sharing DDQN at convergence, increasing from 181.4 (std.dev 4.1) to 454.5 (std.dev 5.9) for quantile SUPER-DDQN measured at 800k training steps. In both Battle and Adversarial-Pursuit, we enabled SUPER-sharing for only one of the two teams each, and see that this significantly improved performance of the sharing team (as measured by sum of rewards of all agents in the team across each episode), especially mid-training. For instance in Battle, the blue team performance increase from -19.0 (std.dev 11.0) for no-sharing DDQN to 5.5 (std.dev 8.7) for quantile SUPER-DDQN at 150k timesteps. In Adversarial-Pursuit, the prey performance increased from -719.8 (std.dev 66.8) to -506.3 (std.dev 35.0) at 300k timesteps.

SUPER performs significantly better than SEAC (blue) and MADDPG (violet). MADDPG performed very poorly on all domains despite extensive attempts at hyperparameter tuning. SEAC shows some learning in Pursuit at 800k timesteps, but remains well below even baseline DDQN performance, again despite extensive hyperparameter tuning. We have also run experiments with SEAC in Pursuit for significantly longer (not shown in the figure), and saw that performance eventually catches up after around 8M timesteps, i.e. a roughly ten-fold difference in speed of convergence. SUPER (red) significantly outperforms parameter-sharing DQN/DDQN (turquoise) in Pursuit and performs similar to it in Battle, whereas parameter-sharing performs significantly better in Adversarial-Pursuit. Note that parameter sharing has often been observed to perform extremely well [40, 39, 11]. We therefore consider this a strong positive result.

Finally, these results and conclusions show remarkable stability across settings, underlying algorithm, and hyperparameters. In Section C.2 in the appendix we show that similar results hold using SUPER

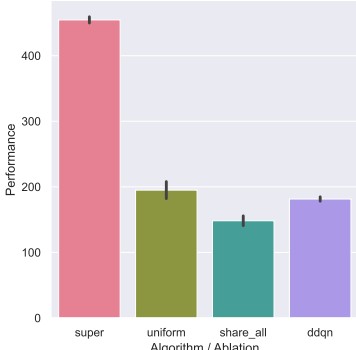

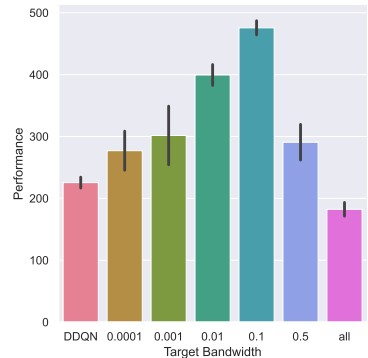

Figure 3: Performance of quantile SUPER vs share-all and uniform random experience sharing in Pursuit at 800k timesteps.

Figure 4: Performance of quantile SUPER with varying bandwidth in Pursuit at 1-2M timesteps.

on a baseline DQN (non-dueling, non-double) algorithm. In Section C.3 in the appendix we show results from SUPER on dueling DDQN on domains where the opposing team agents are co-evolving rather than pretrained. In Section C.5 in the appendix we show that these improvements over baseline DDQN are stable across a wide range of hyperparameter choices. Relative performance results and conclusions in all these cases largely mirror the ones presented here.

## 5.5 Ablations

We ran an ablation study to gain a better understanding of the impact of two key features of our algorithm: selectivity (only sharing a small fraction of experiences) and priority (sharing experiences with the highest td-error). To this end, we compared SUPER against two cut-down versions of the algorithm: One, we share all experiences indiscriminately. This is also equivalent to a single "global" replay buffer shared between all agents. Two, we share (a small fraction of) experiences, but select experiences uniformly at random. Figure 3 shows the performance of these two ablations at end of training in Pursuit and Battle. We see that both ablations perform significantly worse than SUPER, with neither providing a significant performance uplift compared to baseline DDQN.[4] This shows that both selectivity as well as priority are necessary to provide a performance uplift, at least in some domains.

## 5.6 Bandwidth Sensitivity

In the Pursuit domain, we performed an analysis of the performance of SUPER-DDQN for varying target bandwidths ranging from 0.0001 to 1 (sharing all experiences). Figure 4 shows the converged performance of SUPER-DDQN with quantile-based experience selection in Pursuit. A label of "DQN" indicates that no sharing is taking place, i.e. decentralized DDQN; a label of "all" indicates that all experiences are shared, without any prioritized selection. Numerical labels give different target bandwidths. Two things stand out to us: First, sharing all experiences indiscriminately does not result in increased performance. In fact, at convergence, it results in slightly lower performance than no-sharing DDQN. Second, there is a clear peak of performance around a target bandwidth of 0.01 - 0.1, which also holds for gaussian and stochastic experience selection (we refer the reader to the appendix for more details). We conclude that sharing experiences *selectively* is crucial for learning to benefit from it.

## 5.7 Experience Selection

Gaussian experience selection performed similarly to the quantile selection we designed it to approximate. Its actual used bandwidth was however much less responsive to target bandwidth than the other two variants. We believe this demonstrates that in principle approximating the actual distribution of

---

[4]The situation in Adversarial Pursuit is more nuanced and highly dependent on training time, but overall shows a similar picture especially early in training.

td-errors using mean and standard deviation is feasible, but that more work is needed in determining the optimal value of $c$ in equation 3. Stochastic experience selection (dotted red) performs similar or worse than both other variants, but generally still comparably or better than baseline DQN/DDQN.

## 6    Conclusion & Discussion

**Conclusion**    We present selective multiagent PER, a selective experience-sharing mechanism that can improve DQN-family algorithms in multiagent settings. Conceptually, our approach is rooted in the same intuition that Prioritized Experience Replay is based on, which is that td-error is a useful approximation of how much an agent could learn from a particular experience. In addition, we introduce the a second key design choice of selectivity, which allows semi-decentralized learning with small bandwidth, and drastically improves performance in some domains.

Experimental evaluation on DQN and dueling DDQN shows improved performance compared to fully decentralized training (as measured in sample efficiency and/or converged performance) across a range of hyperparameters of the underlying algorithm, and in multiple benchmark domains. We see most consistent performance improvements with SUPER and a target bandwidth of 0.01-0.1 late in training, more consistent than indiscriminate experience sharing. Given that this effect appeared consistently across a wide range of hyperparameters and multiple environments, as well as on both DQN and dueling DDQN, the SUPER approach may be useful as a general-purpose multi-agent RL technique. Equally noteworthy is a significantly improved performance early in training even at very low bandwidths. We consider this to be a potential advantage in future real-world applications of RL where sample efficiency and rapid adaption to new environments are crucial. SUPER consistently and significantly outperforms MADDPG and SEAC, and outperforms parameter sharing in Pursuit (but underperforms in Adversarial-Pursuit, and shows equal performance in Battle).

**Discussion**    Our selective experience approach improves performance of both DQN and dueling DDQN baselines, and does so across a range of environments and hyperparameters. It outperforms state-of-the-art multi-agent RL algorithms, in particular MADDPG and SEAC. The only pairwise comparison that SUPER loses is against parameter sharing in Adversarial-Pursuit, in line with a common observation that in practice parameter sharing often outperforms sophisticated multi-agent RL algorithms. However, we note that parameter sharing is an entirely different, fully centralized training paradigm. Furthermore, parameter sharing is limited in its applicability, and does not work well if agents need to take on different roles or behavior to successfully cooperate. We see this in the Pursuit domain, where parameter sharing performs poorly, and SUPER outperforms it by a large margin. The significantly higher performance than MADDPG and SEAC is somewhat expected given that baseline non-sharing DQN algorithms often show state-of-the-art performance in practice, especially with regard to sample efficiency.

It is noteworthy that deterministic (aka "greedy") experience selection seems to perform slightly better than stochastic experience selection, while in PER the opposite is generally the case [30]. We have two hypotheses for why this is the case. One, we note that in PER, the motivation for stochastic prioritization is to avoid low-error experiences never being sampled (nor re-prioritized) in many draws from the buffer. On the other hand, in SUPER we only ever consider each experience once. Thus, if in stochastic experience selection a high-error experience through random chance is not shared on this one opportunity, it will never be seen by other agents. In a sense, we may prefer deterministic experience selection in SUPER for the same reason we prefer stochastic selection in PER, which is to avoid missing out on potentially valuable experiences. Two, in all our current experiments we used (stochastic) PER when sampling training batches from the replay buffer of each agent. When using stochastic SUPER, each experience therefore must pass through two sampling steps before being shared and trained on by another agent. It is possible that this dilutes the probability of a high-error experience being seen too much.

We would also like to point out a slight subtlety in our theoretical motivation for SUPER: We use the sending agent's td-error as a proxy for the usefulness of an experience for the receiving agent. We believe that this is justified in symmetric settings, and our experimental results support this. However, we stress that this is merely a heuristic, and one which we do not expect to work in entirely asymmetric domains. For future work, we would be interested to explore different experience selection heuristics. As an immediate generalization to a more theoretically grounded approach, we wonder if using the td-error of each (potential) receiving agent could extend SUPER to asymmetric settings, and if it

could further improve performance even in symmetric settings. While this would effectively be a centralized-training approach, if it showed similar performance benefits as we have seen in symmetric settings for SUPER, it could nevertheless be a promising avenue for further work. Beyond this, we would be interested to explore other heuristics for experience selection. For instance, we are curious if the sending agent could learn to approximate each receiver's td-error locally, and thus retain the decentralized-with-communication training capability of our current approach. However, given that td-error is intrinsically linked to current policy and thus highly non-stationary, we expect there would be significant practical challenges to this.

In this current work, we focus on the DQN family of algorithms in this paper. In future work, we would like to explore SUPER in conjunction with other off-policy RL algorithms such as SAC [12, 21] and DDPG [32]. The interplay with experience sampling methods other than PER, such as HER [3] would also be interesting. If the improvements we see in this work hold for other algorithms and domains as well, this could improve multi-agent RL performance in many settings.

Finally, our approach is different from the "centralized training, decentralized execution" baselines we compare against in the sense that it does not require fully centralized training. Rather, it can be implemented in a decentralized fashion with a communications channel between agents. We see that performance improvements scale down even to very low bandwidth, making this feasible even with limited bandwidth. We think of this scheme as "decentralized training with communication" and hope this might inspire other semi-decentralized algorithms. In addition to training, we note that such a "decentralized with communication" approach could potentially be deployed during execution, if agents keep learning. While this is beyond the scope of the current paper, in future work we would like to investigate if this could help when transferring agents to new domains, and in particular with adjusting to a sim-to-real gap. We envision that the type of collaborative, decentralized learning introduced here could have impact in future applications of autonomous agents ranging from disaster response to deep space exploration.

## Acknowledgements

The project was sponsored, in part, by a grant from the Cooperative AI Foundation. The content does not necessarily reflect the position or the policy of the Cooperative AI Foundation and no endorsement should be inferred.

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

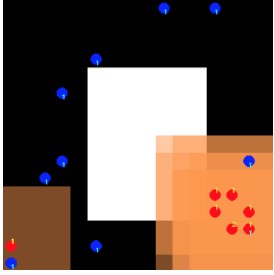

Figure 5: Pursuit Environment

# A  Algorithm Details

---

**Algorithm 1** SUPER algorithm for DQN

---

**for** each training iteration **do**
    Collect a batch of experiences $b$ {DQN}
    **for** each agent $i$ **do**
        Insert $b_i$ into $\text{buffer}_i$ {DQN}
    **end for**
    **for** each agent $i$ **do**
        Select $b_i^* \subseteq b_i$ of experiences to share[1] {SUPER}
        **for** each agent $j \neq i$ **do**
            Insert $b*_i$ into $\text{buffer}_j$ {SUPER}
        **end for**
    **end for**
    **for** each agent $i$ **do**
        Sample a train batch $b_i$ from $\text{buffer}_i$ {DQN}
        Learn on train batch $b_i$ {DQN}
    **end for**
**end for**

[1] See section "Experience Selection"

---

Algorithm 1 shows a full pseudocode listing for SUPER on DQN.

# B  Experiment Domains

## B.1  PettingZoo

**SISL: Pursuit**    is a semi-cooperative environment, where a group of pursuers has to capture a group of evaders in a grid-world with an obstacle. The evaders (blue) move randomly, while the pursuers (red) are controlled by RL agents. If a group of two or more agents fully surround an evader, they each receive a reward, and the evader is removed from the environment. The episode ends when all evaders have been captured, or after 500 steps, whichever is earlier. Pursuers also receive a (very small) reward for being adjacent to an evader (even if the evader is not fully surrounded), and a (small) negative reward each timestep, to incentivize them to complete episodes early. We use 8 pursuers and 30 evaders.

**MAgent: Battle**    is a semi-adversarial environment, where two groups of opposing teams are battling against each other. An agent is rewarded 0.2 points for attacking agents in the opposite team, and 5 points if the other agent is killed. All agents start with 10 health points (HP) and lose 2 HP in each attack received, while regaining 0.1 HP in every turn. Once killed, an agent is removed from the environment. An episode ends when all agents from one team are killed. The action space, of size 21 is identical for all agents, with (8) options to attack, (12) to move and one option to do nothing. Since no additional reward is given for collaborating with other agents in the same team, it is considered to be more challenging to form collaboration between agents in this environment. We use a map of size $18 \times 18$ and 6 agents per team.

**MAgent: Adversarial Pursuit**   is a predator-prey environment, with two types of agents, prey and predator. The predators navigate through obstacles in the map with the purpose of tagging the prey. An agent in the predators team is rewarded 1 point for tagging a prey, while a prey is rewarded $-1$ when being tagged by a predator. Unlike in the Battle environment, prey agents are not removed from the game when being tagged. Note that prey agents are provided only with a negative or zero reward (when manage to avoid attacks), and their aim is thus to evade predator agents. We use 8 prey agents, and 4 predator agents.

### B.2   MeltingPot

We chose a subset of three domains ("substrates") from the MeltingPot package [18]. Cooperative Mining is a collaborative resource harvesting game, where cooperation between players allows them to mine a more valuable type of resource than individual action does. Rationalizable Coordination is a coordination game embedded in a larger rich environment. Similarly Stag Hunt is as stag hunt game payoff matrix embedded in a larger game world.

### B.3   Atari

We also ran experiments on a version of the Atari 2600 game Space Invaders [4, 6]. For this, we created two different variants. In the main variant, we ran the game in a native two-player mode. By default, observations from this are not strictly anonymous, as each player is identified using a fixed color. We thus made two changes to make observations strictly anonymous: First we added a colored bar to the bottom of the observation that indicates which player is being controlled. Then, we randomly shuffle each episode which agent controls which player in the game. This way, both policies learn to control each of the two player colors, and are able to make sense of observations from each other.

In a second variant, we took two separate instances of the game running a single-player mode and linked them to create a two-player environment. This provides an additional "sanity check" to ensure that the modifications we made to the primary variant did not influence results in our favor.

## C   Further Experimental Results

### C.1   Additional Results on DDQN

In addition to the final performance shown in the main text, we show in Table 1 numerical results from all experiments.

Table 1: Performance in all three environments, taken at 800k timesteps (Pursuit), 300k timesteps (Battle), 300k timesteps (Adv. Pursuit). Numbers in parentheses indicate standard deviation. Highest performance in each environment is printed in bold.

| Algorithm \| Env | pursuit | battle | adversarial_pursuit |
|---|---|---|---|
| ddqn | 181.43 (+-4.16) | 9.46 (+-0.86) | -719.78 (+-66.82) |
| ddqn_paramshare | -139.03 (+-121.11) | 15.15 (+-8.64) | **-268.78 (+-60.11)** |
| ddqn_shareall | 148.27 (+-9.22) | 9.25 (+-6.60) | -568.91 (+-40.82) |
| maddpg | -342.24 (+-4.12) | -10.31 (+-0.0) | -1071.71 (+-8.56) |
| seac | 32.51 (+-70.27) | -2.73 (+-0.03) | -993.40 (+-120.22) |
| super_ddqn_gaussian | 373.63 (+-9.15) | 16.28 (+-3.42) | -480.37 (+-15.79) |
| super_ddqn_quantile | **454.56 (+-5.89)** | **17.05 (+-2.74)** | -506.29 (+-35.03) |
| super_ddqn_stochastic | 266.42 (+-85.52) | 7.43 (+-5.64) | -582.59 (+-15.20) |

### C.2   DQN and Dueling DDQN

For all of the DDQN and SUPER-DDQN variants discussed in Section 5, we consider also variants based on standard DQN. Figure 6 shows results from these experiments.

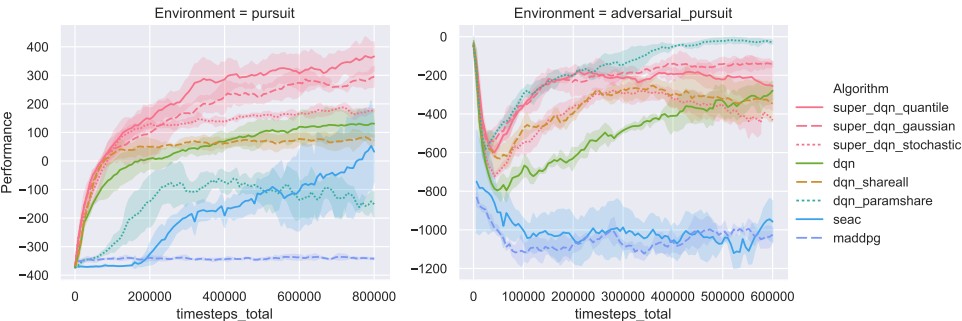

Figure 6: Performance of SUPER-DQN variants with target bandwidth 0.1 on Pursuit and Adversarial-Pursuit. For Pursuit, performance is the total mean episode reward from all agents. For Adversarial-Pursuit, performance is the total mean episode reward from all agents in the prey team. Shaded areas indicate one standard deviation.

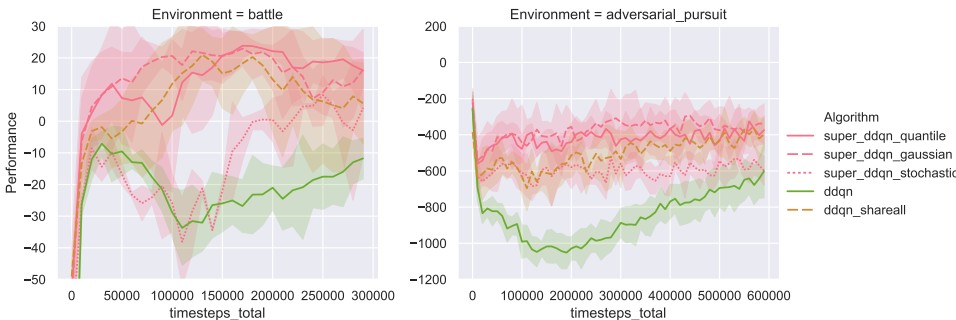

Figure 7: Performance of SUPER-dueling-DDQN variants with target bandwidth 0.1 on all Battle and Adversarial=Pursuit, with co-evolving opponents. Performance is the total mean episode reward from all agents in the sharing team (blue team in Battle, prey team in Adversarial-Pursuit). Shaded areas indicate one standard deviation.

### C.3 Co-Evolving Teams

In Battle and Adversarial-Pursuit we further show a variant where the opposing team are co-evolving with the blue / prey team. In this variant, all agents start from a randomly initialized policy and train concurrently, using a DDQN algorithm. However, only the blue / prey team share experiences using the SUPER mechanism. We only do this for the DDQN baseline as well as discriminate and share-all SUPER variants. This is in part because some of the other baseline algorithms do not support concurrently training opposing agents with a different algorithm in available implementations; and in part because we consider this variant more relevant to real-world scenarios where fully centralized training may not be feasible. We aim to show here how sharing even a small number of experiences changes the learning dynamics versus to non-sharing opponents. Figure 7 shows this variant.

### C.4 Further Ablations

In addition to the ablations presented in the main text, we include here additional results from the Battle environment in Figure 9. Results are broadly similar to the results in the main text, with a notably bad performance for uniform random experience sharing.

### C.5 Stability Across Hyperparameters

Figure 8 shows performance of no-sharing DDQN and SUPER-DDQN for different hyperparameters. As we can see, SUPER-DDQN outperforms no-sharing DDQN consistently across all the hyperparameter settings considered.

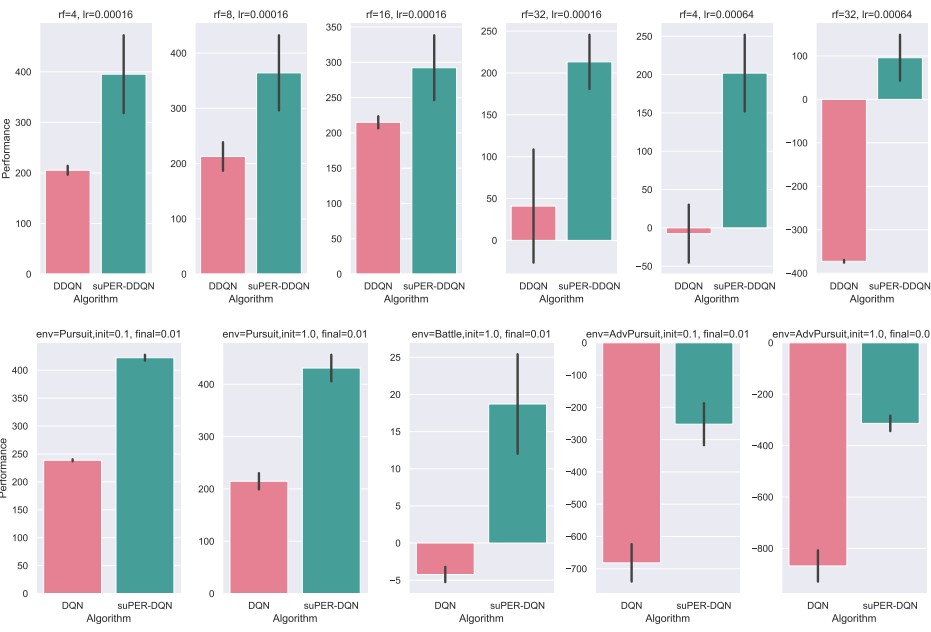

Figure 8: Performance of DDQN and SUPER-DDQN (gaussian experience selection, target bandwidth 0.1) for differing hyperparameter settings of the underlying DDQN algorithm. Top: Different learning rates and rollout fragment lengths in Pursuit. Bottom: Different exploration settings in Pursuit and co-evolving variants of Batle and Adversarial-Pursuit. Hyperparameters otherwise identical to those used in Figure 1. Performance measured at 1M timesteps in Pursuit, 300k timesteps in Battle, 400k timesteps in Adversarial-Pursuit.

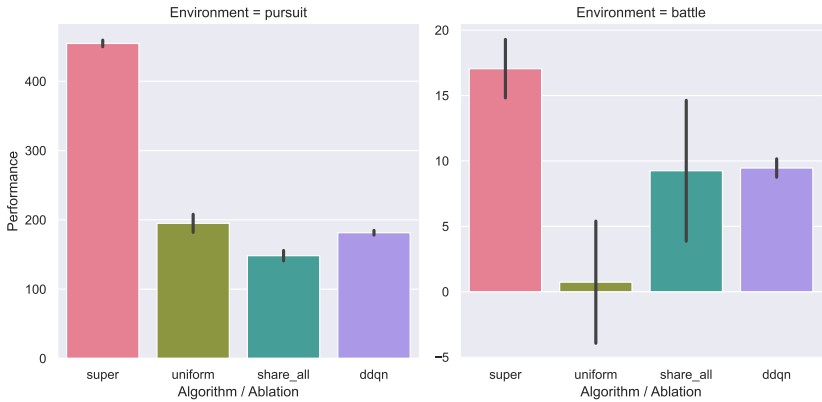

Figure 9: Performance of quantile SUPER vs share-all and uniform random experience sharing in Pursuit at 800k timesteps.

## D   Additional Analysis of Bandwidth Sensitivity

We present here a more detailed analysis of bandwidth sensitivity of SUPER-DDQN in the three experience selection modes we discuss in the main text. Figure 10 shows the mean performance across five seeds for gaussian (left), quantile (middle) and stochastic (right) experience selection, at 1-2M timesteps (top) and at 250k timesteps (bottom). We can see that at 1-2M timesteps and a target bandwidth of $0.1$, all three experience selection criteria perform similary. One thing that stands out is that stochastic selection has much lower performance at other target bandwidths, and also much less performance uplift compared to no-sharing DDQN at 250k timesteps at any bandwidth. Gaussian experience selection appears to be less sensitive to target bandwidth, but upon closer analysis we found that it also was much less responsive in terms of how much actual bandwidth it used at different

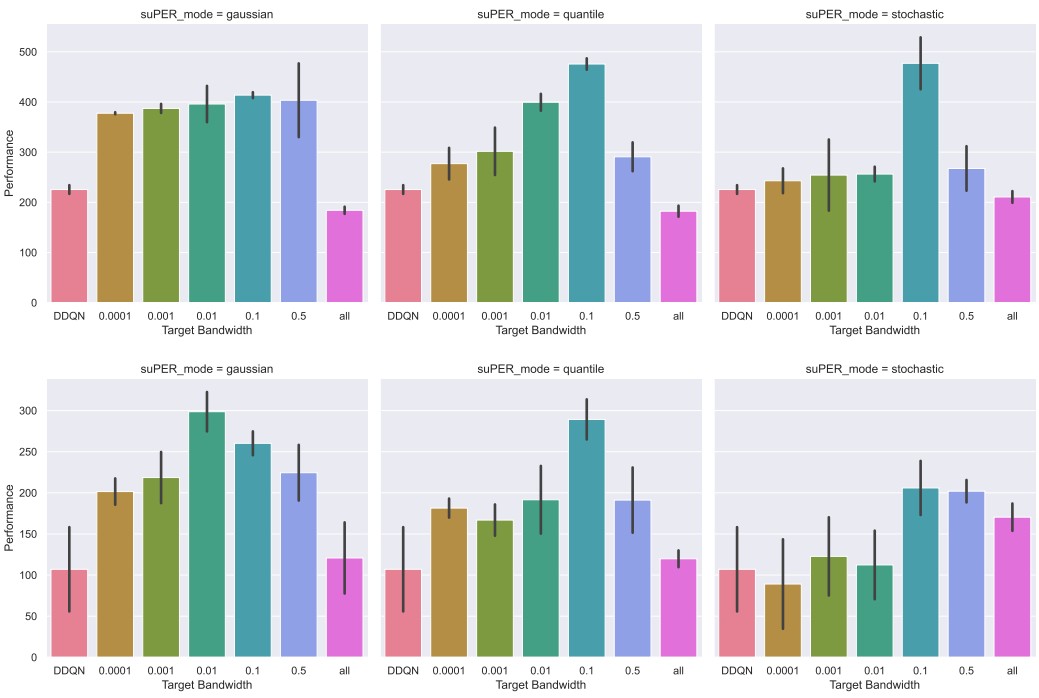

Figure 10: Performance of SUPER with different experience selection and varying bandwidth in Pursuit at 1-2M timesteps (top) and at 250k timesteps (bottom).

settings. Figure 11 (left) shows the actual bandwidth used by each selection criterion at different target bandwidths. We can see that quantile and stochastic experience hit their target bandwidth very well in general.[5] What stands out, however, is that gaussian selection vastly overshoots the target bandwidth at lower settings, never going significantly below 0.01 actual bandwidth.

What is a fairer comparison therefore is to look at performance versus actual bandwidth used for each of the approaches, which we do in Figure 11 (middle, at 1-2M timesteps, and right, at 250k timesteps). For these figures, we did the following: First, for each experience selection approach and target bandwidth, we computed the mean performance and mean actual bandwidth across the five seeds. Then, for each experience selection mode, we plotted these $(\mathrm{mean\,actual\,bandwidth, mean\,performance})$ (one for each target bandwidth) in a line plot.[6] The result gives us a rough estimate of how each approach's performance varies with actual bandwidth used. We see again that stochastic selection shows worse performance than quantile at low bandwidths, and early in training. We also see that gaussian selection very closely approximates quantile selection. Notice that gaussian selection never hits an exact actual bandwidth of 0.1, and so we cannot tell from these data if it would match quantile selection's performance at its peak. However, we can see that at the actual bandwidths that gaussian selection does hit, it shows very similar performance to quantile selection. As stated in the main text, our interpretation of this is that using mean and variance to approximate the exact distribution of absolute td-errors is a reasonable approximation, but that we might need to be more clever in selecting $c$ in equation 3.

---

[5]Quantile selection overshoots at 1e-4 (0.0001) target bandwidth and is closer to 1e-3 actual bandwidth usage, which we attribute to a rounding error, as we ran these experiments with a window size of 1500 (1.5e+3), and a quantile of less than a single element is not well-defined.

[6]Because each data point now has variance in both $x$- and $y$-directions, it is not possible to draw error bars for these.

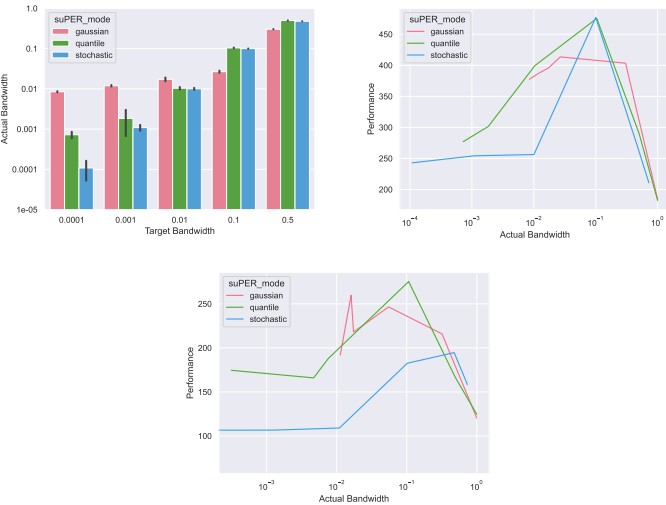

Figure 11: Left: Actual bandwidth used (fraction of experiences shared) at different target bandwidths. Middle, right: Performance compared to actual bandwidth used at 1-2M and 250k timesteps.

## E  Experiment Hyperparameters & Details

We performed all experiments using the open-source library **RLlib** [20]. Experiments in Figure 1 and 6 were ran using RLlib version 2.0.0; experiments in other figures were run using version 1.13.0. Environments used are from PettingZoo [38], including SISL [11] and MAgent [45]. The SUPER algorithm was implemented by modifying RLlib's standard DQN algorithm to perform the SUPER experience sharing between rollout and training. Table 2 lists all the algorithm hyperparameters and environment settings we used for all the experiments. Experiments in the "Stability across hyperparameters" section had hyperparameters set to those listed in Table 2 except those specified in Figure 8. Any parameters not listed were left at their default values. Hyperparameters were tuned using a grid search; some of the combinations tested are also discussed in the "Stability across hyperparameters" section. For DQN, DDQN and their SUPER variants, we found hyperparameters using a grid search on independent DDQN in each environment, and then used those hyperparameters for all DQN/DDQN and SUPER variants in that environment. For all other algorithms we performed a grid search for each algorithm in each environment. For MADDPG we attempted further optimization using the Python **HyperOpt** package [5], however yielding no significant improvement over our manual grid search. For SEAC, we performed a grid search in each environment, but found no better hyperparameters than the default. We found a CNN network architecture using manual experimentation in each environment, and then used this architecture for all algorithms except MADDPG where we used a fully connected net for technical reasons. We tested all other algorithms using both the hand-tuned CNN as well as a fully connected network, and found that the latter performed significantly worse, but still reasonable (and in particular significantly better than MADDPG using the same fully connected network, on all domains).

All experiments were repeated with three seeds. All plots show the mean and standard deviation of these seeds at each point in training. For technical reasons, individual experiment runs did not always report data at identical intervals. For instance, one run might report data when it had sampled 51000 environment timesteps, and another run might report at 53000 environment timesteps. In order to still be able to report a meaningful mean and standard deviation across repeated runs, we rounded down the timesteps reported to the nearest $k$ steps, i.e. taking both the data above to represent each run's performance at 50000 steps. We set $k$ to the target reporting interval in each domain (8000 timesteps in Pursuit, 6000 timesteps in the other two domains). Where a run reported more than once in a 10000 step interval, we took the mean of its reports to represent that run's performance in the interval. Mean and standard deviation were calculated across this mean performance for each of the five seeds. To increase legibility, we applied smoothing to Figures 1 and 6 using an exponential window with $\alpha = 0.3$ for Pursuit, $\alpha = 0.1$ for Battle, and $\alpha = 0.25$ for Adversarial-Pursuit, and $\alpha = 0.3$ for Atari.

No smoothing was found to be necessary for MeltingPot results. This removes some noise from the reported performance, but does not change the relative ordering of any two curves.

# F   Implementation & Reproducibility

All source code is included in the appendix and will be made available on publication under an open-source license. We refer the reader to the included README file, which contains instructions to recreate the experiments discussed in this paper.

Table 2: Hyperparameter Configuration Table - SISL: Pursuit

| **Environment Parameters** | | | |
|---|---|---|---|
| **HyperParameters** | **Value** | **HyperParameters** | **Value** |
| max cycles | 500 | x/y sizes | 16/16 |
| shared reward | False | num evaders | 30 |
| horizon | 500 | n catch | 2 |
| surrounded | True | num agents(pursuers) | 8 |
| tag reward | 0.01 | urgency reward | -0.1 |
| constrained window | 1.0 | catch rewards | 5 |
| obs range | 7 | | |
| **CNN Network** | | | |
| CNN layers | [32,64,64] | Kernel size | [2,2] |
| Strides | 1 | | |
| **SUPER / DQN / DDQN** | | | |
| learning rate | 0.00016 | final exploration epsilon | 0.001 |
| batch size | 32 | nframework | torch |
| prioritized replay_alpha | 0.6 | prioritized replay eps | 1e-06 |
| dueling | True | target network update_freq | 1000 |
| buffer size | 120000 | rollout fragment length | 4 |
| initial exploration epsilon | 0.1 | | |
| **MADDPG** | | | |
| Actor lr | 0.00025 | Critic lr | 0.00025 |
| NN(FC) | [64,64] | tau | 0.015 |
| framework | tensorflow | actor feature reg | 0.001 |
| **SEAC** | | | |
| learning rate | 3e-4 | adam eps | 0.001 |
| batch size | 5 | use gae | False |
| framework | torch | gae lambda | 0.95 |
| entropy coef | 0.01 | value loss coef | 0.5 |
| max grad norm | 0.5 | use proper time limits | True |
| recurrent policy | False | use linear lr decay | False |
| seac coef | 1.0 | num processes | 4 |
| num steps | 5 | | |

Table 3: Hyperparameter Configuration Table- MAgent: Battle

| **Environment Parameters** | | | |
|---|---|---|---|
| **HyperParameters** | **Value** | **HyperParameters** | **Value** |
| minimap mode | False | step reward | -0.005 |
| Num blue agents | 6 | Num red agents | 6 |
| dead penalty | -0.1 | attack penalty | -0.1 |
| attack opponent reward | 0.2 | max cycles | 1000 |
| extra features | False | map size | 18 |

| **CNN Network** | | | |
|---|---|---|---|
| CNN layers | [32,64,64] | Kernel size | [2,2] |
| Strides | 1 | | |

| **SUPER / DQN / DDQN** | | | |
|---|---|---|---|
| learning rate | 1e-4 | batch size | 32 |
| framework | torch | prioritized replay_alpha | 0.6 |
| prioritized replay eps | 1e-06 | horizon | 1000 |
| dueling | True | target network update_freq | 1200 |
| rollout fragment length | 5 | buffer size | 90000 |
| initial exploration epsilon | 0.1 | final exploration epsilon | 0.001 |

| **MADDPG** | | | |
|---|---|---|---|
| Actor lr | 0.00025 | Critic lr | 0.00025 |
| NN(FC) | [64,64] | tau | 0.015 |
| framework | tensorflow | actor feature reg | 0.001 |

| **SEAC** | | | |
|---|---|---|---|
| learning rate | 3e-4 | adam eps | 0.001 |
| batch size | 5 | use gae | False |
| framework | torch | gae lambda | 0.95 |
| entropy coef | 0.01 | value loss coef | 0.5 |
| max grad norm | 0.5 | use proper time limits | True |
| recurrent policy | False | use linear lr decay | False |
| seac coef | 1.0 | num processes | 4 |
| num steps | 5 | | |

Table 4: Hyperparameter Configuration Table - MAgent: Adversarial Pursuit

| **Environment Parameters** | | | |
|---|---|---|---|
| **HyperParameters** | **Value** | **HyperParameters** | **Value** |
| Number predators | 4 | Number preys | 8 |
| minimap mode | False | tag penalty | -0.2 |
| max cycles | 500 | extra features | False |
| map size | 18 | | |

| **Policy Network** | | | |
|---|---|---|---|
| CNN layers | [32,64,64] | Kernel size | [2,2] |
| Strides | 1 | | |

| **SUPER / DQN / DDQN** | | | |
|---|---|---|---|
| learning rate | 1e-4 | batch size | 32 |
| framework | torch | prioritized replay alpha | 0.6 |
| prioritized replay eps | 1e-06 | horizon | 500 |
| dueling | True | target network update_freq | 1200 |
| buffer size | 90000 | rollout fragment length | 5 |
| initial exploration epsilon | 0.1 | final exploration epsilon | 0.001 |

| **MADDPG** | | | |
|---|---|---|---|
| Actor lr | 0.00025 | Critic lr | 0.00025 |
| NN(FC) | [64,64] | tau | 0.015 |
| framework | tensorflow | actor feature reg | 0.001 |

| **SEAC** | | | |
|---|---|---|---|
| learning rate | 3e-4 | adam eps | 0.001 |
| batch size | 5 | use gae | False |
| framework | torch | gae lambda | 0.95 |
| entropy coef | 0.01 | value loss coef | 0.5 |
| max grad norm | 0.5 | use proper time limits | True |
| recurrent policy | False | use linear lr decay | False |
| seac coef | 1.0 | num processes | 4 |
| num steps | 5 | | |

Table 5: Hyperparameter Configuration Table - MeltingPot

**Policy Network**

| CNN layers | [16,128] | Kernel size | [8,8], [sprite_x, sprite_y] |
|---|---|---|---|
| Strides | 8,1 | | |

**SUPER / DDQN**

| learning rate | 0.00016 | batch size | 32 |
|---|---|---|---|
| framework | torch | prioritized replay alpha | 0.6 |
| prioritized replay eps | 1e-06 | horizon | 500 |
| dueling | True | target network update_freq | 1000 |
| buffer size | 30000 | rollout fragment length | 4 |
| initial exploration epsilon | 0.1 | final exploration epsilon | 0.001 |

Table 6: Hyperparameter Configuration Table - Atari

**Policy Network**

| Model Configuration | RLlib Default | | |
|---|---|---|---|

**SUPER / DDQN**

| learning rate | 0.0000625 | batch size | 32 |
|---|---|---|---|
| framework | torch | adam_epsilon | 0.00015 |
| dueling | True | target network update_freq | 8000 |
| buffer size | 100000 | rollout fragment length | 4 |
| initial exploration epsilon | 0.1 | final exploration epsilon | 0.01 |

