# OpenReview forum: "Selectively Sharing Experiences Improves Multi-Agent Reinforcement Learning"
_NeurIPS.cc/2023/Conference — NeurIPS 2023 poster_

### Official Review · Reviewer_3mbW · 2023-06-10

**Soundness:** 4 excellent
**Presentation:** 3 good
**Contribution:** 3 good
**Rating:** 6
**Confidence:** 3

**Summary:**

The paper proposes a modified prioritized experience replay algorithm where agents share a minimal amount of high-quality samples to improve learning.

**Strengths:**

- The proposed method is very simple (therefore easy to implement and reproduce), and widely applicable for many cooperative RL environments, yet the method seems powerful and to significantly improve learning speed.
-Potential to big impact to the general RL community, and wide applicability such as being added to standard RL frameworks,
- Paper is clear and easy to understand.

**Weaknesses:**

- The main weakness of the paper is that the evaluation domain is very simple for an exclusively-empirical paper. I would expect an evaluation in a domain of ~Starcraft complexity to be included.

- I don't understand why the performance of the algorithms was shown just for a couple of arbitrary training points. You should have instead added a line graph where the reader could see the performance of each algorithm/baseline over time.

- The paper touches on it briefly but it would be good to have a deeper discussion of on which multiagent scenarios this idea could be used and how. Is any typpe of "observation translation" needed for the agents to share the experiences, and what happens if the agents are heterogenous.

- Ideally, I would like to have seem a comparison against some Transfer learning methods that use approximately the same amount of transferred data. Many applicable algorithms could be found in the survey:

 Silva, Felipe Leno, and Anna Helena Reali Costa. "A survey on transfer learning for multiagent reinforcement learning systems." Journal of Artificial Intelligence Research 64 (2019): 645-703.

**Questions:**

No specific question, feel free to clarify misunderstandings.

**Limitations:**

no foreseeable negative societal impact.

---

> ### Author Rebuttal · Authors · 2023-08-10
>
> Dear Reviewer 3mbW,
>
> We would like to thank you for your very helpful comments and suggestions. We would like to respond to some of your comments:
>
> * Starcraft: While we agree that Starcraft is an interesting domain, it was designed specifically as a challenge domain for credit assignment in shared reward settings, which is not the problem our work aims to solve. This would therefore not be a useful benchmark domain for our work. However, based on your feedback, we have run additional experiments on several more complex domains from the Melting Pot package, and include results in the PDF shared with all reviewers. We see a strong performance uplift for SUPER compared to baseline DDQN in all tested environments, in some cases even greater than in our earlier experiments.
> * Line vs bar plots: We had line plots in earlier working versions of our paper, but received clear feedback from readers that they were difficult to decipher due to the large number of algorithms we run in each domain, and that a simpler plot or table would communicate our results more effectively. However, we still include the full learning curves in the supplementary material should you wish to have a look. We will also attempt to further improve the formatting of these, or perhaps to split them into different figures for comparison to different baselines. If we find a way to make them more readable, we will move them to the main text.
> * Heterogenous agents: No translation is needed for homogenous agents / anonymous environments. We feel that anonymity is not a very strong limitation, as in particular the real world and the laws of physics are anonymous, i.e. our work could be applied to real-world robotics applications. Note that our idea does work for heterogenous behavior, and we see this in the Pursuit domain. For non-anonymous settings, our idea could extend in a relatively straightforward manner to settings with different reward functions, by estimating the recipient’s reward using their Q-function (while still learning about environment dynamics from the received experience). Settings where agents observe and act in entirely different ways are more difficult; it might be possible to develop an approach similar to ours using more abstract semantics, but this would be outside the scope of our current work. We will clarify this in the paper text.
> * Thank you for your suggestion regarding transfer learning. We will study the suggested paper carefully to see if any direct comparisons are possible, and we will in any case include a discussion of transfer learning and citation of the survey paper in the next revision of our paper.
>
> We thank you again for your constructive feedback, and would welcome further discussion in case you have any remaining concerns.

---

> > ### Comment · Reviewer_3mbW · 2023-08-11
> > **Post-rebuttal**
> >
> > Starcraft was just an example. The domains added to the experimental evaluation and too simple for a empirically-focused paper on Neurips and this is the main reason why my score was "borderline accept".
> >
> > While Starcraft might not be the best option there are plenty of more complex environments such as HOF, Robocup 3D or 2D simulation, robotics simulators, multiplayer video games and many others. You would be better than anyone to identify which domain is the best to evaluate your approach but to be honest super simplistic environments that don't require any "clever engineering" to work as the ones added in the experimental evaluation speak loudly as method that is hard to broadly apply in different scenarios. I know it's not the case for your approach but this is definitely a first impression for a reader skimming through your paper for deciding if it's worth their time to read the paper in detail.

---

> > > ### Author Response · Authors · 2023-08-13
> > > **Clarifications regarding domain complexity**
> > >
> > > Dear Reviewer 3mbW,
> > >
> > > Thank you for your prompt engagement. We apologize in case our rebuttal came across in the wrong way, we did not mean to dismiss your feedback. Rather the opposite, we meant to agree and believe we addressed your suggestions. We believe that there must be a misunderstanding about the domains we use, which are state-of-the-art multi-agent RL benchmark domains, and we would like to convince you that our current experiments in fact already are at the level of complexity you are asking us to target.
> > >
> > > Complexity of an environment can of course be measured in many ways, but by all metrics we can think of we would consider the domains we use to be at least equal to the domains you suggest. Consider for instance the following:
> > >
> > > - Dimensionality: Melting Pot features high-dimensional image observations, which agents need to learn to extract relevant features from. In contrast, StarCraft Multi-Agent Challenge (SMAC) gives agents a feature vector observation that already contains the relevant features. The same is true for Half-Field Offense (HFO), as well as other RoboCup and Robotics simulators we are aware of. Melting Pot is more comparable to some videogames like Atari in this regard (some Melting Pot environments have a higher dimensionality than Atari). Action and state spaces seem to be similar across all the domains mentioned, with perhaps the main notable difference being some robotics domains using continuous control (which also puts them out of scope for our current DQN-focused work).
> > > - World dynamics: Robotics simulators, HFO and Robocup typically have a physics engine. On the other hand, Melting Pot has dynamics like resource exhaustion; these may be simpler, but also less predictable / more stochastic. SMAC is based on a complex game engine, but only uses a small restricted subset of this; in each scenario, it’s primarily movement and health of agents that change; this is very similar to what happens in Melting Pot.
> > > - Domain skills: Robocup and some videogames require the agent to learn trajectory estimation; on the other hand our domains, similarly to Starcraft, require predicting the behavior of opponent agents controlled by a complex and non-linear policy, arguably a more difficult task. In SMAC agents merely select an enemy unit to target, whereas in some of our domains agents have to capture a target by literally catching up to it.
> > > - Coordination skills: In Starcraft, agents have to learn one of several “micro-trick” skills for coordination with other agents, depending on the scenario, such as positioning or alternating fire. Some of our experiments require very similar coordination, such as surrounding a moving target from multiple directions, or collaborating to harvest a resource.
> > > - Agent interaction: Crucially, some domains feature very loose agent interactions, such as in some videogames - e.g. in Atari Space Invaders, no cooperation between players is needed, and in fact each player could complete the game by themselves. Some other domains require coordination between players; This includes StarCraft and Robocup, but also all of our domains. One aspect in which some of our domains, especially the newly added Melting Pot environments are significantly more complex than many other domains is in their incentive structure: Whereas many other domains are either purely cooperative (e.g. Starcraft or Robocup within each team), or zero-sum (SC and RC across teams), Melting Pot was designed to feature complex incentive and payoff structures, such as coordination games embedded inside the environment.
> > >
> > > Overall, we believe that the environments we already have in the paper are at the level of complexity you are asking us for. We would like to stress the final point regarding payoff structures in particular - We suspect that one reason why the domains we use might appear simple is because they are purpose-made for RL, rather than taken from a human-centered domain (like videogames). However, this is a fallacy - these domains are challenging because they were made to be challenging for multi-agent RL, more so than prior domains could be. This is part of what makes them especially suited for evaluating our work.
> > >
> > > Does the above satisfy you of the complexity of our experimental evaluation? If not, is there a specific aspect of domain complexity where you feel other domains would make evaluation of our work more convincing? We would want to further strengthen our paper, and it would be helpful if we knew what specifically to look out for when choosing additional domains.
> > >
> > > We would also like to thank you for your candid feedback, we had not realized that the complexity of our experiments is not at all obvious in our paper and will discuss these in greater length in the paper. Beyond discussing the points raised above, is there anything else we could do to more clearly communicate the level of complexity of our experiments?

---

> > > > ### Comment · Reviewer_3mbW · 2023-08-13
> > > >
> > > > While melting pot is a more complex environment, the evaluation was only added post-rebuttal (if I understood correctly), which limits the ability of cross-examining the experiments now with the limited time reviewers have during the discussion period.
> > > >
> > > > One way in which predator-prey is less complex than the ideal (even if you make it more complicated with input as images and some stochastic effect actions)  is that it is pretty obvious to generalize the few states in which the predators have to coordinate to capture the prey.
> > > >
> > > > Personally, I am not a fan of overly-engineered domains such as melting pot and particles, they are just toy problems you have to run for longer. I prefer domains that were not primarily built for training AI agents, such as games or robotic simulators.
> > > >
> > > > Either way, I am increasing my score a bit trusting that the melting pot experiment will be appropriately included and discussed in the paper.

---

> > > > > ### Author Response · Authors · 2023-08-19
> > > > >
> > > > > Dear Reviewer 3mbW,
> > > > >
> > > > > We would like to thank you for your continued engagement, and for taking our additional experiments and explanations into considerations. We also thank you for your additional feedback and comments. We appreciate your point regarding domains not built for AI training. Following this, we have run additional experiments on a videogame, specifically the Atari videogame Space Invaders. We again see a significant performance uplift of SUPER over DDQN, e.g. at 5M timesteps, SUPER achieves 59.98 (+- 1.07) clipped reward, compared to DDQN at 47.46 (+-5.23). Furthermore, this is robust across several slight variations we have attempted (primarily related to how we process observations to make them anonymous). We aren’t able to update the PDF at this stage, but would be happy to discuss any questions you might have about the experiments, and will of course add these in the next revision of our paper. With this additional domain, would your concern about experiment complexity be fully addressed?
> > > > >
> > > > > Again, we thank you for your constructive feedback, which has been very helpful in further strengthening our paper.

---

### Official Review · Reviewer_mQNR · 2023-06-21

**Soundness:** 4 excellent
**Presentation:** 3 good
**Contribution:** 3 good
**Rating:** 6
**Confidence:** 5

**Summary:**

The paper proposes a heuristic for sharing the experience between agents in Multi-Agent Reinforcement Learning (MARL) setting. It is a modification of Prioritised Experience Replay (PER) method, where the agents only share experiences, which td-error exceeds a certain threshold (paper proposes several ways to choose the threshold). The experimental evaluation show that the proposed approach delivers better performance against a set of reasonable baselines.

**Strengths:**

- The paper is clear and well positioned within the related work
- The approach is simple, elegant and easy to reproduce.
- Experiments are clean and well presented
- Experimental results are convincing

Overall I believe that the paper can have a moderate, yet significant impact on MARL. Due to it's simplicity the method could be easily deployed and (if the results generalise beyond the 3 domains presented) provide a very decent "bang for the buck" in terms of performance vs compute/implementation complexity.

**Weaknesses:**

- Somewhat limited choice of environments. For example, what happens in environments that require division of labour, say overcooked? I would love to see the performance of the proposed method on a broader benchmark, for example MeltingPot [1]. I understand that compute is a limitation and not all of the environments are appropriate (anonymous), but maybe a subset, which is and requires division of labour would be a great test.
- Only applicable to anonymous MARL (same reward, actions and observations). This is a big limitation, especially the reward.
- The contribution 4. (line 43) is confusing and would suggest authors remove it. The "emergent communication" is a very different problem and I don't think discussing it here is helpful.

Overall, there are some clear limitations to the approach (listed above). My main concern is whether the improvements generalise beyond the presented domains, especially to the domains where division of labour / diversity of policy is required for the population of agents to be successful.

1. Leibo, J. Z., Dueñez-Guzman, E. A., Vezhnevets, A., Agapiou, J. P., Sunehag, P., Koster, R., ... & Graepel, T. (2021, July). Scalable evaluation of multi-agent reinforcement learning with melting pot. In International conference on machine learning (pp. 6187-6199). PMLR.

**Questions:**

How could the system be upgraded to move beyond "anonymous" MARL?

**Limitations:**

The authors are honest about limitations of their method and it is appreciated.

---

> ### Author Rebuttal · Authors · 2023-08-10
>
> Dear Reviewer mQNR,
>
> We would like to thank you for your insightful comments and suggestions. We would like to clarify a few details:
>
> * Division of labor:
>    * Regarding the Overcooked domain in particular, while we agree that this is an exciting domain, the sparsity of reward make this an extremely challenging domain. The Melting Pot paper itself shows that baseline RL fails to show any non-zero returns during training, i.e. when not paired with a scenario agent. (Figure 4 in Leibo et al, “collaborative cooking” subfigures, S-P column, where all entries are 0.) In initial experiments we ran, we similarly failed to get any significant learning using DDQN in this domain. Our understanding is that the Overcooked domain is more aimed at testing behavior vis-a-vis different paired partners (e.g. few-shot coordination, generalization between partner behaviors or adapting to human partners), and less suitable for testing agents learning concurrently from scratch.
>    * However, note that we already see division of labor in the Pursuit domain. Here multiple agents have to surround a prey from different directions. Our hypothesis is that part of why SUPER outperforms parameter sharing on this domain is that the SUPER agents are able to learn diversified strategies.
> * Anonymous MARL: Note that reward is not a strict limitation, as one could estimate the recipient’s reward using the recipient’s Q-function, and still benefit from the environment dynamics seen in the sender’s experience. See also our response to reviewer aJ1o’s first question. Furthermore, we would like to suggest that anonymity is not as big a limitation as one would think at first glance. In particular, the real world is anonymous! What we have in mind as a prototypical application of our work is a group of robots exploring a new environment or task in the real world together. For instance, our work could lead to self-driving cars sharing a small subset of the real-world data they collect with one another. We believe such real-world applications are manifold enough to warrant a paper focus on them (and it is our decentralized-with-communication paradigm that makes this approaches especially well-suited to future real-world deployment).
> * Contrib. 4: Thank you for this useful feedback, upon re-reading this we agree. We will remove this.
>
> Regarding your question, as per above, our approach already demonstrates positive results in a domain with some division of labor, and it would be possible to generalize it to domains with heterogeneous rewards.
>
> In addition, we thank you for your suggestion of the Melting Pot project. We have run experiments on a number of environments from the Melting Pot package, and include results in the PDF shared with all reviewers. We see a significant performance improvement for SUPER compared to baseline DDQN across all the domains we tested, in some cases even greater in magnitude than in our original experiments.
>
> Finally, we would like to thank you for your comment on easy deployment of our method, which we also value highly. We designed our code to drop into the standard RLlib package with minimal modifications, and if accepted plan to submit a pull request to include SUPER in the official RLlib package, so as to make it easily accessible to the wider community.
>
> We thank you again for your constructive feedback, and would welcome further discussion in case you have any remaining concerns.

---

> > ### Author Response · Authors · 2023-08-19
> >
> > Dear Reviewer mQNR,
> >
> > In addition to the new Melting Pot experiments, we have now also run experiments on the Atari videogame Space Invaders. We again see a significant performance improvement when using SUPER over DDQN, e.g. at 5M timesteps, SUPER achieves 59.98 (+- 1.07) clipped reward, compared to DDQN at 47.46 (+-5.23).. We will add these experiments to the next revision of our paper, and believe they further strengthen the experimental evaluation of our approach. We would greatly appreciate your feedback as to whether this addresses your concerns, as well as about the clarifications and Melting Pot experiments added in the original rebuttal, as we want to make sure the next iteration of our paper fully addresses your feedback.. We again would like to thank you for all your constructive feedback, which has helped strengthen our paper further.

---

### Official Review · Reviewer_8wH6 · 2023-06-29

**Soundness:** 3 good
**Presentation:** 3 good
**Contribution:** 2 fair
**Rating:** 3
**Confidence:** 5

**Summary:**

The paper proposes SUPER, a communication-based cooperative multi-agent reinforcement learning method with semi-decentralized training. In SUPER, each agent relays its highest-td-error experiences to the other agents, who insert them directly into their replay buffer, which they use for learning. The author believes that the selective exchange of trajectories between agents can improve the performance of basic off-policy algorithms in cooperative multi-agent tasks.

**Strengths:**

1. The paper is well organized and easy to follow.
2. The motivation is clear, that is, agents selectively relay some important experiences to other agents to help them learn faster.
3. The basic algorithm of SUPER is independent off-policy methods. Therefore, it relaxes the assumption of centralized training.
4. In the experimental section, authors carry out many ablation experiments, which show the superior performance of SUPER in given scenarios.
5. Key details of the implement are sufficiently well-described and key resources are available.

**Weaknesses:**

1. SUPER is a communication-based method. However, the baselines in the experimental section do not include a communication-related algorithm.
2. There are too few experimental environments to demonstrate the generalization of SUPER in different domains. In addition, the action spaces of environments in this paper are all discrete .
3. Authors only compared the performance between the vanilla DQN-like algorithms and their SUPER variants, which means the SUPER framework was not applied to other off-policy methods such as DDPG.
4. Authors did not actually quantify the bandwidth cost brought by the communication between agents, which is an important measure.

**Questions:**

1. Does SUPER perform better than other classical communication methods? And How about their bandwidth costs?
2. The author claims that SUPER is a method based on semi-decentralized training, so how much less is its bandwidth cost compared to the centralized training algorithm?
3. What about the performance of the SUPER variant of other off-policy methods such as DDPG? Does it outperform MADDPG in given environments?
4. Trying to evaluate the performance of SUPER in more domains will greatly improve the convincingness of experiments.

**Limitations:**

Authors discussed the limitations of their work in detail in the last section.

---

> ### Author Rebuttal · Authors · 2023-08-10
>
> Dear Reviewer 8wH6,
>
> We would like to thank you for your constructive feedback and suggestions. Regarding your questions:
>
> 1. Could you clarify what algorithms you have in mind here? We are not aware of any other work that uses communication as part of its learning algorithm in a similar way; the closest would be SEAC, which we already compare against. Note that what we do is very different from algorithms that learn to communicate (e.g. work on emergent communication, or the “social influence” paper on sequential social dilemmas); we are not sure what a meaningful comparison to such papers could look like.
>
> 2. We explore this in Section 5.6 and particularly in Figure 3. To be clear, centralized training would use a bandwidth of (at least) 1. For SUPER, we see highest performance at a bandwidth of 0.1, or a 10-fold reduction; performance at 0.01 is still almost as high, at a 100-fold reduction compared to centralized training. Even at 0.001, or a 1000-fold reduction, we see significantly higher performance than centralized baselines.
>
> 3. Regarding DDPG, the single-agent version of DDPG works exclusively for continuous-control applications. For the MADDPG paper, this was modified using a Gumbel-Softmax relaxation to make it compatible with discrete action spaces, but this is not standard and recent research has shown evidence that this might be problematic (see e.g. Tilbury et al, Revisiting the Gumbel-Softmax in MADDPG, arXiv:2302.11793). We therefore do not believe that benchmarks using DDPG on discrete domains would provide a reliable baseline to compare against. On the other hand, for continuous actions, we are not aware of any relevant widely-used multi-agent benchmark domains that use continuous control, nor of any published tuned baselines. Available implementations of MADDPG and SEAC also seem to focus on discrete actions and it is unclear to us how well they support continuous control (if at all). Given this, it would be difficult to perform meaningful comparisons.
>
>    * To clarify, we mention off-policy algorithms as a suggestion for future work; we did not intend to claim that our paper applies SUPER to every off-policy algorithm. We will move the sentence "Beyond DQN and its variants, SUPER could in principle apply to any off-policy RL algorithm." to Discussion and clearly label it as a future direction. Applying a new method to one algorithm is relatively common. For instance, the MADDPG paper makes a general point about actor-critic approaches but focuses on one particular algorithm, and SEAC could be applied to any actor-critic algorithm, but is developed in detail only for A2C.
>
>    * However, note that we do in fact run benchmarks on two different off-policy algorithms, DQN and dueling DDQN, the former in the supplementary material. While for historical reasons DQN-type algorithms tend to be named as variants of DQN (unlike policy-gradient based methods), these are very different algorithms: Double DQN and dueling DQN are as different from DQN as actor-critic PPO and A2C are from plain policy gradient. (Double DQN can be seen as a form of actor-critic architecture, and dueling DQN learns an advantage function like A2C does.) In other words, we already show SUPER on two substantially different off-policy algorithms.
>
> 4. Following your feedback, we have run additional experiments on a number of environments from the Melting Pot package. We consistently see a significant performance uplift of SUPER compared to baseline DDQN. In fact, the effect is even larger in some of these additional domains than it was in our original experiments.
>
> We thank you again for your constructive feedback, and would welcome further discussion in case you have any remaining concerns.

---

> > ### Comment · Reviewer_8wH6 · 2023-08-13
> > **Thank you for your reply.**
> >
> > I sincerely appreciate the author's response to the questions I raised. However, there are still some issues that have not been addressed.
> >
> > 1. While the author asserts that SUPER is very different from algorithms that learn to communicate, in reality, it is also a method of communication, albeit transmitting information in the form of trajectories. There is a significant body of work related to communication [1-3].
> >
> > 2. Most importantly, it is unclear how the communication cost of SUPER compares to other communication methods. Intuitively, in pixel-based tasks, the communication cost among agents in SUPER appears to be higher compared to other communication methods.
> >
> > 3. Both DQN and DDQN are similar in that they are off-policy value-based methods. I am curious about whether this approach could also be applied to off-policy policy-based methods.
> >
> > 4. I greatly appreciate the additional experiments the authors have conducted. However, I believe the current experimental environment remains relatively simplistic.
> >
> > The contributions presented in the paper are not substantial enough to meet the standards for acceptance at NeurIPS. Therefore, I am maintaining my initial score.
> >
> > **Reference**
> >
> > [1] Peng, Peng et al. Multiagent Bidirectionally-Coordinated Nets: Emergence of Human-level Coordination in Learning to Play StarCraft Combat Games. 2017.
> >
> > [2] Kim, Daewoo et al. Learning to Schedule Communication in Multi-agent Reinforcement Learning. 2019.
> >
> > [3] Jiang, Jiechuan and Zongqing Lu. Learning Attentional Communication for Multi-Agent Cooperation. 2018.

---

> > > ### Author Response · Authors · 2023-08-13
> > > **Clarification regaring communication approaches and experimental evaluation**
> > >
> > > Dear Reviewer 8wH6,
> > >
> > > Thank you for your further comments. We would like to clarify regarding communication approaches and our experimental evaluation.
> > >
> > > Regarding **communication approaches**: we stress that learning-to-communicate approaches are not doing communication in a way that is comparable to what we do. In SUPER, the learning algorithm uses communication during *training*, whereas approaches like [2] and [3] learn to have the agent policy communicate during *execution*. For this reason, a direct quantitative comparison would not be meaningful. They also solve different problems: SUPER addresses a learning problem by leveraging experiences collected by other agents during training. Learn-to-communicate approaches solve domain problems by using communication to coordinate among multiple strategies during execution.
> > >
> > > You are correct that depending on the observation dimensionality and configured bandwidth, SUPER might use more bandwidth during training than communication approaches might during execution. A direct comparison would only make sense however in situations where there is no distinction between training and deployment, which we do not believe is common. In such settings, a quantitative comparison would be entirely dependent on the specifics of each environment and algorithm hyperparameters, and we do not think it would be possible to draw any general conclusions. It would also be entirely possible to combine SUPER and a learning-to-communicate approach, so we believe that if anything, an ablation study would make more sense than a benchmark comparison. (I.e., it would be interesting to see if “communication+SUPER” provide more or less of a performance uplift over “communication-without-SUPER”, than “no-communication+SUPER” does over “no-communication-without-SUPER”, rather than comparing “SUPER” vs “communication”.) However, we believe that this would be firmly outside the scope of our current work. It is also worth noting that SUPER can be entirely opportunistic in its use of communication, i.e. use whatever is available, including at varying rates, and loss of communication would not pose any problems (it would just mean temporarily falling back to standard DQN). So even if in some scenario where such a comparison was meaningful, a communication approach were to give better performance-per-bandwidth, there would still be reasons to consider SUPER over or in combination with such an approach.
> > >
> > > Regarding [1], this is a centralized-training/decentralized-execution approach similar to MADDPG, and is more readily comparable to SUPER. Being a CTDE approach this would incur a communication cost of at least 1.0 (experiences transmitted per timestep), as each transition needs to be communicated to a central learner. Because of this, what we wrote previously about SUPER scaling down to at least 1/1000 the bandwidth holds.
> > >
> > > All this being said, your point is well taken and we greatly appreciate the feedback. We will include a discussion regarding the differences and possible comparisons between SUPER and learning-to-communicate approaches, including all the above points, in the next iteration of the paper. Does this address your concern?
> > >
> > >
> > >
> > > Regarding **experiments**: We are somewhat confused by this feedback, as Melting Pot is a state-of-the-art benchmark collection, by a leading research lab in this area, designed specifically to present challenging multi-agent benchmarks. We wonder if there might have been a misunderstanding about the domains we use? The new Melting Pot experiments have high-dimensional image observations, difficult coordination skills, complex world dynamics, and challenging incentive structures. Please see also our detailed reply to reviewer 3mbW, where we outline why we consider them among the hardest multi-agent benchmarks. Does our explanation there address your concerns? If not, could you elaborate what types of experiments you would find convincing? We would like to strengthen our paper, but these are the most advanced applicable domains we are aware of.

---

> > > > ### Author Response · Authors · 2023-08-19
> > > >
> > > > Dear Reviewer 8wH6,
> > > >
> > > > In addition to the new Melting Pot experiments, we have now also run experiments on the Atari videogame Space Invaders. We again see a significant performance improvement when using SUPER over DDQN, e.g. at 5M timesteps, SUPER achieves 59.98 (+- 1.07) clipped reward, compared to DDQN at 47.46 (+-5.23). We will add these experiments to the next revision of our paper, and believe they further strengthen the experimental evaluation of our approach. We would greatly appreciate your feedback as to whether this addresses your concerns, as well as about the clarifications in our previous message, as we want to make sure the next iteration of our paper fully addresses your feedback. We again would like to thank you for all your constructive feedback, which has helped strengthen our paper further.

---

### Official Review · Reviewer_aJ1o · 2023-07-05

**Soundness:** 2 fair
**Presentation:** 3 good
**Contribution:** 2 fair
**Rating:** 4
**Confidence:** 4

**Summary:**

This paper introduces a semi-centralized multi-agent reinforcement learning algorithm named SUPER, by incorporating limited communication in the training phase.
With minor modifications, agents are able to achieve better performance than the baseline algorithms by sharing a limited amount of experiences.


**Strengths:**

1. This paper is easy to understand.
2. The proposed algorithm is easy to implement and demonstrates superior performance compared to the baselines.
3. The idea of sharing important experiences seems good.

**Weaknesses:**

1. The difference between SUPER and previous works is not clearly clarified.
As mentioned in the paper that one of the previous works use importance sampling to integrate experiences, it might be more persuasive to compare the performance with experiences.
2. The evaluated environments might be too simple in multi-agent reinforcement learning area.
As far as my knowledge, it's more common to observe only a team reward during training, while individual rewards are rare situations.
3. The experimental comparison with multi-agent baselines might be unfair.
It seems that in DQN and SUPER experiments agents receive their individual rewards, while MADDPG, as far as I know, takes team rewards, which misleads agents.

**Questions:**

1. Intuitively, it seems that sharing only works for isomorphic agents. What about heterogeneous agents?
2. What if the agents are not provided with their adequate indivudal rewards?
3. What if we incorporate importance sampling on the shared experience?

**Limitations:**

Yes, they have.

---

> ### Author Rebuttal · Authors · 2023-08-10
>
> Dear Reviewer aJ1o,
>
> We would like to thank you for your helpful comments and suggestions, and would like to offer a few clarifications:
> * Difference to previous work: To clarify, while SEAC uses importance sampling, this is only used to “translate” action probabilities for off-policy experiences. However, there is no sampling going on in the sense that only a subset of experiences are shared - all experiences are shared in SEAC. A key innovation in our work is the selectivity and sharing only a small number of important experiences. It is this difference that enables us to develop a decentralized algorithm that needs only limited communication, a training paradigm that is much more broadly applicable than previous centralized algorithms.
> * Training environments: To our knowledge, it is not the case that shared-reward environments are more common than individual-reward environments, both types of environments just provide different challenges (and so are perhaps common in different sub-communities). Shared-reward primarily leads to a credit assignment problem (see e.g. literature on value decomposition networks), whereas individual-reward leads to questions about incentives and cooperation. A large number of seminal works in multi-agent RL has focused on individual-reward environments, e.g. work on game-playing including Poker, the breakthrough AlphaGo AI, and the two recent Science papers on Stratego and Diplomacy; papers on learning game-theoretic equilibria (e.g. V-learning for Nash, recent work on Stackelberg equilibria); work on reward-shaping; sequential social dilemmas; as well as a broad line of literature on opponent shaping (e.g. LOLA, M-FOS, etc). Shared-reward environments are less appropriate for our work, as our algorithm does not aim to solve the credit assignment problem.
> * Comparisons: This is a misunderstanding, MADDPG does take individual rewards. The original MADDPG paper used a mix of cooperative and competitive environments. In the experiments in our paper, MADDPG uses individual rewards for a fair comparison.
>
> To answer your questions:
> 1. This depends on the type of heterogeneity: Heterogeneous behavior is something SUPER can work with, and we already see this in the Pursuit domain. Heterogeneous rewards would also be relatively easy to incorporate, by estimating the recipient’s reward using their own Q-function (this way agents would be able to learn from each other about environment dynamics, even if they have different reward functions). For agents that perceive and act in entirely different spaces, learning from each others' experiences is more difficult, of course. However, it might be possible to do something similar using some form of higher-level semantics.
> 2. As above, shared-reward environments are primarily a credit assignment problem, which is not what we are trying to solve. That said, we would like to draw attention to the fact that the Pursuit domain already does partial shared reward: Multiple agents have to surround a prey, and then all those agents share the reward for doing so. Our experiments show that SUPER works well in this setting. Additionally, it might be possible to combine SUPER with credit assignment techniques such as value-decomposition networks; however we feel that trying to do this in the same paper as the base SUPER algorithm would detract from the key messages we would like to convey.
> 3. Importance sampling is only needed to translate off-policy experiences for on-policy policy-gradient methods. It is not necessary to do this for DQN algorithms (nor even possible, as the “log-pi” term that needs correcting in the policy gradient theorem is simply not present in the q-learning updates).
>
> We appreciate your point regarding environments we use in our experiments. Based on this feedback, we have run experiments in several more complex domains from the Melting Pot package, and include results in the PDF shared with all reviewers. We see a significant performance improvement for SUPER compared to baseline DDQN across all the domains we tested, in some cases even greater in magnitude than in our original experiments.
>
> We thank you again for your constructive feedback, and would welcome further discussion in case you have any remaining concerns.

---

> > ### Author Response · Authors · 2023-08-19
> >
> > Dear Reviewer aJ1o,
> >
> > In addition to the new Melting Pot experiments, we have now also run experiments on the Atari videogame Space Invaders. We again see a significant performance improvement when using SUPER over DDQN, e.g. at 5M timesteps, SUPER achieves 59.98 (+- 1.07) clipped reward, compared to DDQN at 47.46 (+-5.23). We will add these experiments to the next revision of our paper, and believe they further strengthen the experimental evaluation of our approach. We would greatly appreciate your feedback as to whether this addresses your concerns, as well as about the clarifications and Melting Pot experiments added in the original rebuttal, as we want to make sure the next iteration of our paper fully addresses your feedback. We again would like to thank you for all your constructive feedback, which has helped strengthen our paper further.

---

### Author Rebuttal · Authors · 2023-08-10

Dear Reviewers,

In addition to individual responses, we are uploading a PDF with figures from testing SUPER in additional domains, as this was a common point raised in the reviews. Following the suggestion from reviewer mQNR, we used the Melting Pot package for this, and compared SUPER-DDQN against a DDQN baseline on several environments from this package. These are more complex domains (e.g. using image observations) than the environments used in our previous experiments. In all of them, we see a clear performance uplift when using SUPER, in some cases even greater in relative magnitude than in our original experiments.

Due to the short time, we have only been able to run SUPER and a DDQN baseline. However, note that this is the main comparison of interest, as all other baselines use fully centralized training. We believe the new results demonstrate that the performance uplift from using SUPER holds across a range of domains.

---

### Decision · Program_Chairs · 2023-09-21

**Decision:**

Accept (poster)

**Comment:**

This paper describes a way of sharing experiences between agents using replay buffers. Reviewers appreciated the simplicity of the algorithm and the many different environments where it was evaluated, including the larger environments like Melting Pot and Atari which were added during the rebuttal period. There was some discussion of whether the comparison baselines were appropriate with two reviewers questioning this, but I found the authors' responses to be convincing. I also noted that another reviewer commented positively about the experiments.

Hopefully the authors will be able to add the new results they obtained on melting pot and atari during the rebuttal period to the final version of the paper.